# Flavonoid Composition and Pharmacological Properties of *Elaeis guineensis* Jacq. Leaf Extracts: A Systematic Review

**DOI:** 10.3390/ph14100961

**Published:** 2021-09-24

**Authors:** Wai-Kit Tow, Asly Poh-Tze Goh, Usha Sundralingam, Uma Devi Palanisamy, Yasodha Sivasothy

**Affiliations:** 1School of Pharmacy, Monash University Malaysia, Jalan Lagoon Selatan, Bandar Sunway 47500, Selangor, Malaysia; Tow.WaiKit@monash.edu (W.-K.T.); aslygoh.pohtze@monash.edu (A.P.-T.G.); usha.sundralingam@monash.edu (U.S.); 2Jeffrey Cheah School of Medicine and Health Sciences, Monash University Malaysia, Jalan Lagoon Selatan, Bandar Sunway 47500, Selangor, Malaysia; umadevi.palanisamy@monash.edu; 3Monash-Industry Palm Oil Education and Research, Monash University Malaysia, Jalan Lagoon Selatan, Bandar Sunway 47500, Selangor, Malaysia; 4Tropical Medicine and Biology Platform, School of Science, Monash University Malaysia, Jalan Lagoon Selatan, Bandar Sunway 47500, Selangor, Malaysia

**Keywords:** Arecaceae, *Elaeis guineensis* Jacq, oil palm leaves, flavonoids, pharmacological activity

## Abstract

The oil palm tree (*Elaeis guineensis* Jacq.) originates from West and Central Africa, and it is cultivated in Malaysia for its oil-producing fruits. Malaysia is the world’s second largest palm oil producer and the world’s largest exporter to date. Consequently, the Malaysian oil palm industry constantly generates a huge amount of biomass with the major contributor being the leaves. A large percentage of these leaves remain underutilized, making them a promising source of raw materials that can be converted into value-added products. The present review summarizes and discusses the flavonoid composition, total phenolic and flavonoid content, and the in vitro and in vivo pharmacological properties exhibited by the extracts of the leaves of *E. guineensis*. The purpose of this systematic review is to highlight the potential of valorizing the leaf extracts of the oil palm tree as pharmaceutical and cosmeceutical agents.

## 1. Introduction

The oil palm trees (*Elaeis*) are tropical perennial plants consisting of two species of the family Arecaceae. The African oil palm, *Elaeis guineensis* Jacq. is indigenous to West Africa, while the American oil palm, *Elaeis oleifera* (Kunth) Cortés, is native to tropical Central America and South America. Both species are used in commercial agriculture in the production of palm oil. *E. guineensis* is the principal source of palm oil while *E. oleifera* is only cultivated for local palm oil production [1,2].

Various parts of the *E. guineensis* tree are widely used as traditional medicine in West Africa [3,4]. The leaves are used to promote wound healing. The leaves are squeezed, and the juice that is obtained is placed on wounds to promote healing. The sap of this plant is also used as a laxative, and the partially fermented palm wine is administered to nursing mothers to improve lactation [5,6]. Soap prepared with ash from the fruit husk is used to treat skin infections. A root decoction is used in Nigeria to relieve headaches. The pulverized roots are added to drinks to treat gonorrhea, menorrhagia, and as a cure for bronchitis [5,6]. The fruit mesocarp oil and palm kernel oil are administered as a poison antidote and used externally with several other herbs as a lotion to treat skin diseases. Palm kernel oil is applied to convulsant children to regulate their body temperature. Folk remedies of oil palm include treatment for cancer, headache, and rheumatism as well as use as an aphrodisiac, diuretic, and liniment [5,6].

The oil palm tree is widely cultivated in Malaysia for its oil-producing fruits [2]. Malaysia is the world’s second largest palm oil producer and the world’s largest exporter to date [7]. However, the production of palm oil is associated with a large amount of oil palm biomass (OPB) with the oil palm leaves (OPL) being the most abundant biomass produced by the oil palm industry during harvesting, pruning, and replanting, accounting for more than 50% of the total OPB dry weight [8]. Owing to this, one of the major challenges faced by the oil palm industry is OPL overload. This problem burdens the operators with disposal difficulties and subsequently escalates the disposal operating costs. Alternatively, these OPLs are either left to decompose in the plantation fields or incinerated, which eventually releases harmful gases into the atmosphere, thereby constituting environmental hazards, leading to detrimental effects on the health of humans and animals [9]. In order to circumvent the challenges posed by the improper disposal of the OPLs, research has been carried out to convert the OPLs into renewable energy and value-added products with varied applications, hence the move toward sustainable palm oil production [10]. Owing to their high fiber and nutrient composition, the OPLs are utilized as animal feed, in the production of pulp and paper, wood-based construction materials, fillers in polymer composites, and renewable sources for carbon and ethanol production. Despite these uses, the OPLs are still considered underutilized [7,8,11,12,13,14].

The present systematic review aims to provide insight into the therapeutic potential of the OPLs being used as a source of bioactive secondary metabolites that can be further developed into pharmaceutical and cosmeceutical products. Thus far, there has been no such study on this commercially and medicinally important crop. This study, which reports on the pharmacological activities of the *E. guineensis* leaf extracts and its biologically active flavonoid composition, will serve as a chemical database for future research while providing researchers with a framework for potential future studies.

## 2. Methods

This systematic review was written based on the guideline requirements of the Preferred Reporting Items for Systematic reviews and Meta-Analyses (PRISMA) [15]. The data extraction and data synthesis were performed independently by the authors (W.-K.T.) and (A.P.-T.G.) based on PICO component (Population; Intervention; Comparison; Outcome) [16].

### 2.1. Search Strategy

After identifying the research objectives, a search was conducted across four electronic databases to identify relevant studies. The four databases selected were Ovid via Embase; Ovid MEDLINE; SciFinder; and Elsevier via Embase. Keywords related to (“*Elaeis guineensis*” or “oil palm leaf” or “oil palm frond”) and (“phytochemical” or “phytonutrient”) were entered into each of the databases. The search terms for the latter were further expanded by including various classes of phenolic compounds, as listed in Table 1. All the terms were used in both of their singular and plural forms. The search was limited to title and abstract. The search results were limited to articles published between January 2001 and February 2020.

### 2.2. Article Eligibility Criteria

As described by the Louisiana State University, inclusion and exclusion criteria constitute a defining process that helps in limiting the evidence synthesis [17]. Articles were selected based on the types of phenolic compounds that have been identified in the OPL and the biological activities exhibited by the OPL extracts. Only scientific research papers published in English over the last 20 years (2001–2020) that report on in vitro and in vivo studies were included in the present systematic review. Scientific research papers related to human trials, clinical trials, and randomized control trials, along with literature reviews, systematic reviews, meta-analysis, patents, and conference proceedings were excluded.

Using the inclusion and exclusion criteria listed above, abstract, title, and full-text screening were carried out independently by two reviewers (W.-K.T. and A.P.-T.G.) using the Covidence software (https://www.covidence.org/) accessed on 31 January 2021. Instances of discordance were resolved by a third senior reviewer (U.S) to prevent biases.

## 3. Results

The search yielded a total of 1242 articles from all four electronic databases. A total of 438 articles remained after the duplicates were removed from Endnote. Then, the retrieved articles in Endnote were uploaded onto Covidence. The duplicates were removed again, by Covidence, with 426 articles remaining. Thirty-seven articles remained after the title and abstract screening was conducted. Full-text screening was subsequently carried out, and the reasons for the exclusion of certain articles are depicted in (Figure 1). Ultimately, 14 articles were selected for qualitative synthesis. Two articles were included as secondary references, resulting in 16 articles for qualitative synthesis. Among these articles, eight described the identification of the secondary metabolites in the OPLE, while seven reported on the TPC of the OPLE. However, only four of the 15 articles analyzed the TFC of the OPLE. These results have been summarized in Table 2.

### 3.1. Secondary Metabolites in the Leaves of E. guineensis

Two different groups of researchers carried out qualitative analyses to determine the types of phytochemicals present in the methanolic OPLE. Results from both analyses indicated the presence of flavonoids [5,23]. In another study conducted by Soundararajan et al. (2012), the FTIR analysis of the methanolic OPLE revealed absorption bands of functional groups characteristic of flavonoids [4,24].

### 3.2. Classification of Flavonoids Found in the Leave of E. guineensis

Flavonoids are polyphenolic compounds known to exhibit numerous health-promoting effects to mankind owing to their powerful antioxidant properties. The structure of flavonoids consists of a 15-carbon skeleton (C6-C3-C6 system) comprising of two aromatic rings (A and B) connected by a three-carbon chain that forms an oxygenated heterocyclic C ring (Figure 2) [25,26,27].

Flavonoids are divided into different subclasses based on the position at which ring B is connected to ring C and the degree of unsaturation and oxidation of ring C. In OPL, the subclasses of flavonoids include flavan-3-ols and flavones (Figure 2). For flavan-3-ols, a hydroxyl group is present at position C3 of its ring C. As for flavones, its ring C has an unsaturation between positions C2 and C3 and a carbonyl group at position C4 [25,27,28].

Flavonoids are present as aglycones, glycosides, methylated and prenylated derivatives. Flavonoid glycosides are the main form of flavonoids in plants. The general glycosylation patterns include *O*-glycoside and *C*-glycoside. The *O*-glycoside has a sugar moiety connected to the hydroxyl group of the aglycone, while the *C*-glycoside is formed through a linkage of the sugar moiety to the aglycone via a carbon–carbon bond [29,30].

#### 3.2.1. Flavan-3-ols

According to the study conducted by Jaffri et al. in 2011, the HPLC analysis of the methanolic OPLE revealed that a noteworthy amount of the flavonoids in the OPL consist of catechins and their gallate esters, in particular (−)-epigallocatechin (EGC, 0.08%), (+)-catechin (0.30%), (−)-epicatechin (EC, 0.01%), (−)-epigallocatechin gallate (EGCG, 0.28%), and (−)-epicatechin gallate (ECG, 0.05%) (Figure 3). These compounds are a group of flavan-3-ols commonly found in green tea [19,28,31,32]. The structure of these catechins in the leaves of *E. guineensis* is characterized by either a hydroxyl or an ester group at position C3 of their ring C, a 3′4′-dihydroxyl or a 3′4′5′-trihydroxyl substituted ring B, and a 5,7-dihydroxyl substituted ring A [33].

#### 3.2.2. Flavones

One of the studies identified luteolin (2.6 µmol/g of dry extract) as the only flavone present in the methanolic OPLE using HPLC [18] (Figure 4). Structurally, luteolin (5,7,3′,4′-tetrahydroxyflavone) has four hydroxyl groups, which are attached at the C5, C7, C3′, and C4′ positions of the flavone backbone [34]. The linoleic acid emulsion test of luteolin exhibited its ability to inhibit (76%) linoleic acid oxidation.

##### Flavone-*C*-glycosides

UHPLC combined with LCMS/MS analyses conducted by Che et al. (2020) on various solvent extracts of OPL (hexane, ethyl acetate, ethyl acetate-methanol, absolute methanol, and aqueous methanol) and its flavonoid-enriched fractions confirmed the presence of ten flavone-*C*-glycosides [21]. They were identified to be derivatives of apigenin and luteolin, including vitexin (apigenin-8-*C*-glucoside), isovitexin (apigenin-6-*C*-glucoside), orientin (luteolin-8-*C*-glucoside), isoorientin (luteolin-6-*C*-glucoside), luteolin-6-8-di-*C*-hexose, apigenin-6,8-di-*C*-hexose, apigenin-6-*C*-pentose-8-*C*-hexose, apigenin-6-*C*-hexose-8-*C*-pentose, luteolin-6-*C*-hexose-8-*C*-deoxyhexose, and apigenin-6-*C*-hexose-8-*C*-deoxyhexose (Figure 5) [21]. Vitexin, isovitexin, orientin, and isoorientin are mono-*C*-glycosylflavones with the anomeric carbon of their sugar moieties (β-D-glucopyranose) being bonded to either the C-8 or C-6 position of the flavone skeleton (apigenin and luteolin) through a carbon–carbon bond [35,36]. A higher concentration of these flavone-*C*-glycosides was found to be present in the extracts of the polar solvents as compared to those of the intermediate and non-polar solvents. The flavonoid-enriched fractions contained very high concentrations of these compounds [21]. However, the same group of researchers in a separate study were only able to identify four flavone-*C*-glycosides, orientin, isoorientin, vitexin, and isovitexin along with two flavon-3-ols, (+)-catechin and (−)-epicatechin, when they analyzed the chemical composition of the OPLE prepared from either absolute methanol or as admixtures with ethyl acetate or water using the ^1^H NMR metabolomics approach [22].

### 3.3. Total Phenolic Content of E. guineensis Leaf Extract

Seven studies discussed in the current review employed the Folin–Ciocalteu method to measure the TPC of the OPLE. The results were derived from a calibration curve constructed from gallic acid and expressed in milligrams of Gallic Acid Equivalents per gram of extract or sample (mg GAE g^−1^ extract or sample) [37] (Table 2). However, the TPC values vary from one study to another. The variation in the phenolic content could be attributed to the geographical location from where the OPLs were collected, the maturity of the OPLs at the time of harvest, the growing season of the oil palm tree, horticultural practices, postharvest storage conditions and the method used to prepare the OPLEs [38,39,40].

Interestingly, the methanolic OPLE was reported to contain 24.3 mg GAE g^−1^ dry weight of non-toxic, antioxidative phenolic compounds at a much higher level than that of green tea (22.5 mg GAE g^−1^ dry weight) [18,19]. Two separate studies conducted by Soundararajan et al. (2012) and Ibraheem et al. (2012) also reported that the OPLs are a source of phenolic compounds.

Han et al. (2010) compared the TPC values of wet and dried OPLE [20]. The phenolic content of the OPLE prepared from dried leaves was found to be two times higher compared to the phenolic content of the OPLE prepared from fresh leaves. Dry conditions are essential to prevent the formation of artifacts as a result of microbial fermentation/growth and subsequent degradation of the plant secondary metabolites.

In an investigation by Hui et al. (2017), the methanol and water extracts had higher TPC values compared to those of the ethyl acetate and hexane extracts [2]. Due to the presence of hydroxyl groups, phenolic compounds are hydrophilic and therefore can dissolve more efficiently in methanol and water. The hydroxyl groups in the phenolic compounds form hydrogen bonds with the hydroxyl groups of methanol and water, thus increasing their solubility in these solvents [41].

A group of researchers investigated the effect of employing different extraction techniques on the TPC values [11]. Significant differences were observed in the TPC values when phenolic compounds were extracted from the OPLs using four different techniques: (i) extraction with ethanol, OPAL M1; (ii) extraction with hexane followed by ethanol, OPAL M2; (iii) extraction with hexane and ethanolic hydrochloric acid, OPAL M3; and (iv) aqueous extraction at 100 °C, OPAL M4. Extraction conducted in an acidic medium or at a very high temperature resulted in significantly lower TPC values compared to when extraction was conducted in a neutral medium or at low or room temperature. Phenolic compounds are known to be unstable at low pH values or at high temperatures and therefore tend to degrade under these conditions during the extraction procedures [11,42,43].

Recently, Che et al. (2020) found that there was a correlation between the TPC values of the OPLEs and the extracting solvents i.e., the value of the TPC increases as the polarity of the extracting solvent increases [22]. The study showed that the aqueous methanol extract had the highest TPC, followed by the absolute methanol and ethyl acetate–methanol extracts. The ethyl acetate extract on the other hand had the lowest TPC. Thus, the study concluded that the methanolic solvent systems are more efficient for extracting phenolic compounds from the OPLs [22].

### 3.4. Total Flavonoid Content of E. guineensis Leaf Extract

Che et al. (2020), Ahmad et al. (2018), and Ibraheem et al. (2012) further analyzed and reported on the quantification of flavonoids in the OPLs using the aluminum chloride colorimetric method [11,22,23]. The standards used in these studies to construct the calibration curve were either quercetin, (+)-catechin, or (−)-epicatechin. TFC values were stated either as milligrams of quercetin equivalents per gram of extract (mg QCE g^−1^ extract) or milligrams of (+)-catechin equivalents per gram of extract (mg CE g^−1^ extract) or milligrams of (−)-epicatechin per gram sample (mg ECE g^−1^ sample). All three investigations indicated that flavonoids were present in the OPLs, and their TFC values have been tabulated in Table 2.

### 3.5. Pharmacological Activities of E. guineensis Leaf Extracts

The alcoholic and aqueous OPLE have been scientifically proven to possess numerous health benefits as they have been reported to exhibit a broad spectrum of pharmacological activities among which include antioxidant, wound healing, hypoglycemic, vascular relaxation, hypocholesterolemic, neuroprotective, neurogenesis, phytoestrogenic, osteogenic, fungicidal, antimicrobial, antihypertensive, cytotoxicity, and toxicity [2,4,5,11,14,18,19,20,21,22,23,31,44,45,46,47]. The pharmacological activities that will be discussed in the following sections have been summarized in Appendix A. A total of 27 in vitro assays were carried out, among which included antioxidant activities (*n* = 21), wound-healing properties (*n* = 3), cytotoxicity effects (*n* = 1), and fungicidal activities (*n* = 2). In vivo studies were categorized by their pharmacological properties including antihypertensive and cardiovascular effects, vascular relaxation, fungicidal activity, antimicrobial activity, toxicity effects, hypocholesterolemic activity, hypoglycemic activity, neuroprotective, neurogenesis effects, phytoestrogenic properties, estrogenic activity, and osteogenic activity (n = 1), wound-healing activity (*n* = 2), and antioxidant activity (*n* = 4).

#### 3.5.1. Antioxidant Activity

The antioxidant potential of the OPLEs was evaluated using both in vitro and in vivo assays: namely, DPPH free radical scavenging activity (*n* = 6), FRAP (*n* = 3), HPSA (*n* = 2), LPO inhibition (*n* = 2), NOS activity (*n* = 2), in vitro LDL oxidation (*n* = 1), XOI activity (*n* = 1), TBARS (*n* = 1), LDL receptor (*n* = 1), *beta*-carotene bleaching (*n* = 1), inhibition of copper-mediated oxidation of LDL, and linoleic acid emulsion system (*n* = 1).

##### In Vitro Antioxidant Activity

DPPH Free Radical Scavenging Activity

In 2012, Ibraheem et al. compared the DPPH free radical scavenging potential of the ethanolic OPLE with that of the positive controls: BHT, a synthetic antioxidant, and vitamin C, a natural antioxidant. The results revealed that the scavenging potential of the OPLE was concentration dependent and was found to be a weaker free radical scavenger as compared to both of the controls. The effective doses of BHT and Vitamin C were respectively three and two-folds more potent compared to the extract itself (Appendix A) [23]. When Soundarajan et al. (2012) investigated the free radical scavenging capacity of the aqueous methanolic OPLE at 1 mg/mL, they found that its scavenging potential was more effective compared to vitamin E but slightly weaker than BHT (Appendix A) [4].

Several studies have proven a correlation between the free radical scavenging activity of OPL extracts with their phenolic content, the polarity of the solvents used, as well as the method of extraction. Che et al. (2020) deduced that as the polarity of the extracting solvent increases, the corresponding increase in the TPC values of the OPLE resulted in an increase in their DPPH free radical scavenging potentials [22]. In this study, the methanolic extracts exhibited significantly greater scavenging activities in comparison with the ethyl acetate extract, which correlated with the trend exhibited by their TPC values (Appendix A) [22]. Similar observations were also made by Hui et al. (2017), whereby the stronger free radical scavenging activity of the methanolic OPLE was consistent with its higher TPC value, while the low phenolic content of the hexane extract resulted in the hexane extract being a weaker free radical scavenger (Appendix A) [2].

Meanwhile, a study by Ahmad et al. (2018) revealed that OPLE prepared in neutral conditions at low or room temperature exhibited stronger free radical scavenging activity compared to the extracts prepared in acidic medium or at a very high temperature. These findings are in accordance with the trend of their TPC values (Appendix A) [11].

Ferric Reducing/Antioxidant Power

In an investigation conducted by Irine et al. in 2003, the methanolic OPLE was found to possess a weak ability to reduce ferric ion compared to green tea and various flavonoid test compounds (Appendix A) [18]. On the other hand, when compared to several other aqueous methanolic plant extracts such as green chili, papaya shoot, and lemongrass, OPL was reported to exhibit significantly stronger ferric ion-reducing activity (Appendix A) [44].

An investigation by Che et al. (2020) revealed that when the polarity of the extracting solvent that was used to prepare the OPLE increases, the corresponding increase in the phenolic content of the OPLE subsequently increases the ferric ion-reducing potential of the OPLE. Once again, the methanolic extracts exhibited remarkably greater activities in comparison with the ethyl acetate extracts (Appendix A) [22].

Hydrogen Peroxide Scavenging Activity

According to Soundararajan et al. (2012), the aqueous methanolic OPLE scavenged hydrogen peroxide in a dose-dependent manner, and its scavenging potential was comparable to that of vitamin C (Appendix A) [4]. However, another study showed that the ethanolic OPLE possessed a lower H_2_O_2_ scavenging efficacy when compared to BHT and vitamin C (Appendix A) [23].

Lipid Peroxidation Activity

Based on the study conducted by Ibraheem et al. (2012), it was observed that the ethanolic OPLE demonstrated weaker anti lipid peroxidation activity as compared to vitamin C (Appendix A) [23]. Similarly, Hui et al. (2017) evaluated the LPO activities of OPLE prepared from methanol, water, ethyl acetate, and hexane [2]. The ethyl acetate extract was found to be the strongest inhibitor among all four extracts. However, the study showed that the ethyl acetate extract was a less effective inhibitor compared to BHT (Appendix A). The polar water extract in contrast exhibited extremely weak LPO inhibition. The inactivity of the water extract could have resulted from the extremely polar nature of the compounds present in the extract which have low solubility in the lipophilic system. Therefore, this hinders contact between the polar compounds and the less polar polyunsaturated fatty acids [2].

Nitric Oxide Scavenging Activity

A study revealed that the methanolic OPLE at 1000 µg/mL was a weaker NO scavenger when compared with ascorbic acid (Appendix A) [4]. The investigation conducted by Che et al. once again confirmed that the polarity of the extracting solvents used to prepare the various OPLE played a role in determining the scavenging potentials of the extracts. The low IC_50_ values of the absolute methanolic and aqueous methanolic OPLE emphasized the efficiency of methanol for the extraction and recovery of antioxidant principles, particularly the phenolic content from the OPL (Appendix A) [22].

In Vitro LDL Oxidation

In a study conducted by Salleh et al. (2002), the inhibition of copper-mediated LDL oxidation by twelve edible plant extracts rich in polyphenols was evaluated (Appendix A). The antioxidant activity i.e., the inhibition of LDL oxidation was determined by measuring the formation of conjugated dienes (lag time) and thiobarbituric acid reactive substances (TBARS). The lag time measures the period in which LDL is protected from copper-mediated oxidation by antioxidants. All of the plant extracts were reported to delay the formation of conjugated dienes by varying degrees. However, only five of the plant extracts were able to inhibit LDL oxidation with the most effective being observed for betel leaf followed by cashew, Japanese mint, semambu, and OPL (Appendix A) [45]. Additionally, these five plant extracts were also found to be successful in inducing the inhibition of TBARS within an hour of its formation upon the oxidation of LDL (Appendix A) [45].

Abeywardena et al. (2002) investigated the in vitro copper-mediated LDL oxidation by the aqueous methanolic extracts prepared from OPL, papaya shoots, green chili, and lemon grass [44]. The extracts of the OPL, papaya shoots, and green chili caused a greater increase in lag time (*p* < 0.05) compared to the vehicle control (55.2 ± 0.5 min) with OPLE being the most effective inhibitor (77.9 ± 0.6 min) [44].

Xanthine Oxidase Inhibitory Activity

Soundararajan et al. (2012) examined the antioxidant activity of methanolic OPLE using the xanthine oxidase inhibitory activity (XOI) assay [4]. At 100 µg/mL, the methanolic OPLE was a weaker inhibitor of the xanthine oxidase enzyme compared to the positive control, allopurinol (Appendix A).

Beta-Carotene–Linoleic Acid Bleaching Assay

Hui et al. (2017) investigated the capacity of various OPLE to inhibit the bleaching of β-carotene. The trend of the results was similar to those obtained when the LPO-inhibiting potentials of the same extracts were assessed (Appendix A) [2].

Regarding the linoleic acid emulsion antioxidant test conducted by Irine et al. (2003), all flavonoid test compounds and OPLE exhibited significantly lower inhibition compared to α-tocopherol (Appendix A). Interestingly, the inhibition potentials of the chloroform, acetone, and methanolic OPLE were comparable to those of (+)-catechin, (−)-epicatechin, and quercetin (Appendix A) [18].

In Vivo Antioxidant Activity

Rosalina Tan et al. (2011) investigated the in vivo antioxidant effects of the ethanolic OPLE in diabetic (DC) male Sprague–Dawley rats (Appendix A) [46]. Hyperglycemic conditions in the DC rats resulted in a decrease in the activities of the endogenous antioxidant enzymes (SOD and CAT) and in the levels of total protein, creatinine, and serum triglycerides (TG) compared to the normal (NC) rats. On the other hand, increases in the concentration of TBARS, in the levels of AST and ALT, and in the percentage of kidney damage, liver necrosis, and inflamed cells were observed in the DC rats. The dose-independent treatment of the DC rats with OPLE was found to be effective (*p* < 0.05) in restoring the activities of the antioxidant, liver, and kidney markers to normal or near-normal levels. Furthermore, the DC rats when treated with the different doses of the ethanolic OPLE resulted in reduced kidney damage, liver necrotic areas, and inflamed cells. These findings indicated that the ethanolic OPLE demonstrated no adverse or chronic toxicity effects on the liver and kidney functions in both the DC and NC rats at the tested concentrations, therefore suggesting that treatment with the OPLE has the potential to exert antioxidative protective effects against damages in the kidney and liver due to oxidative stress conditions [46].

Mohamed et al. (2013) investigated the in vivo antioxidant potential of aqueous ethanolic OPLE on brain SOD activity, CAT activity, GPx activity, and MDA levels in NO-deficient male Wistar Kyoto rats (Appendix A) [19]. The NO deficiency caused a significant reduction (*p* < 0.001) in the MDA levels of these rats, which was accompanied by a decrease in their SOD and CAT activities. The study showed that the dietary OPLE increased the brain SOD and CAT activities as well as the MDA levels in NO-deficient rats. It is noteworthy to mention that the OPLE was more effective than captopril, the positive control, in upregulating the CAT activity to near-normal levels. Treatment with OPLE did not show any effect in the activities of the GPx [19].

In a separate study conducted by Irine et al. (2003), the effects of 0.2% OPLE and 1% oil palm leaf powder (OPLP) on blood CAT and GPx activities and MDA levels of hypercholesterolemic-induced male New Zealand white rabbits were tested for 16 weeks (Appendix A) [18]. The study revealed that at the end of the administration period (week 16) with either 0.2% OPLE or 1% OPLP, the CAT activity and MDA levels in the blood of the hypercholesterolemic induced rabbits were restored to near normal values. Treatment with OPLE did not show any effect in the activities of the GPx [18].

#### 3.5.2. Hypoglycemic Activity

Tan et al., in 2011, evaluated the anti-diabetic potential of the ethanolic extract of the OPL on normal (NC) and diabetic (DC) rats by monitoring its glycemic-modulating effect (Appendix A). The dose-dependent treatment of the ethanolic OPLE significantly (*p* < 0.05) reduced the blood glucose levels in the DC rats. Moreover, the OPLE did not exert any hypoglycemic effects in the NC rats [46].

Weight loss is one of the characteristics of diabetes that results from the defect of glucose metabolism and the excessive breakdown of tissue protein despite the increase in food consumption [48]. Based on the findings of Tan et al. (2011), the untreated DC rats recorded significant (*p* < 0.05) weight loss by about 40% when compared to the NC rats (Appendix A). On the other hand, the DC rats that were dose-dependently administered with the OPLE did not experience any weight loss. The OPLE was found to significantly (*p* < 0.05) prevent weight loss in the DC rats, and the optimum dose was 100 mg/kg body weight [46].

The OPLE treatment also significantly reduced the mortality rate of the DC rats (Appendix A). When treated with either a dosage of 100 or 200 mg/kg body weight, these rats showed lower mortality rates than the untreated DC rats. At 100 and 200 mg/kg body weight, the mortality improved by 71% compared to the untreated DC group [46].

#### 3.5.3. Wound-Healing Activity

In vitro studies on the wound-healing properties of OPLE were carried out by Che et al. (2020) in two separate studies [21,22]. In their first investigation, various solvent extracts of the OPL (hexane, ethyl acetate, ethyl acetate–methanol, absolute methanol, and aqueous methanol), the flavonoid enriched fraction of the aqueous methanolic OPLE, the flavonoid *C*-glycoside-enriched fraction of the aqueous methanolic OPLE, and the purified flavonoid *C*-glycosides (orientin, isoorientin, vitexin, and isovitexin) were evaluated for their efficacy in enhancing the proliferation and migration of the 3T3 fibroblast cells by employing the scratch assay (Appendix A). The absolute methanolic OPLE, aqueous methanolic OPLE, the flavonoid-enriched fraction, flavonoid *C*-glycoside-enriched fraction, and orientin at a 3.125 μg/mL concentration were found to possess effective wound-healing properties. These samples enabled the 3T3 cells to proliferate and migrate rapidly (>90%). As for isoorientin, vitexin, and isovitexin, these compounds facilitated effective proliferation and migration of the 3T3 cells (87–89%) at a lower concentration of 1.563 μg/mL [21]. The high (70–160%) cell proliferation activity of the OPL extracts, the flavonoid-enriched fraction, the flavonoid *C*-glycoside-enriched fraction, and the purified flavonoid *C*-glycosides as determined by the CCK-8 assay, in turn, promoted the growth of the 3T3 cells without having any effect on their cell viability and cellular activity at all tested concentrations (1.563 to 12.5 μg/mL) [21].

Che et al. (2020)’s subsequent study investigated the relationship between the polarity of the extracting solvents used to prepare the OPLE with their phenolic content and antioxidant (DPPH, FRAP, NO) activities (Appendix A). As the polarity of the extracting solvent increased, the increase in the TPC values and the antioxidant activities of the respective OPLE resulted in a corresponding increase in their wound-healing properties [22].

In 2012, Sasidharan et al. studied the in vivo wound healing activity of the OPLE. Full-thickness wounds made on the shaved dorsal area of Sprague–Dawley strain albino rats were treated topically with 10% formulated crude OPLE, while the control rats were treated with only soft yellow paraffin. The decrease in wound diameters during the healing process was monitored for 25 days (Appendix A). Overall, the rats treated with the OPLE had a faster wound closure rate compared to the control group. A significant difference in wound closure was observed between the two groups from day 4 onwards. In later days, the rate of wound closure in the treated group was much higher (*p* < 0.05) than that of the control group. Complete wound closure was observed in the group treated with the OPLE on day 16, whereas it took about 25 days for the wound to close in the control group [5,47].

#### 3.5.4. Vascular Relaxation

Abeywardena et al. (2002) conducted an in vivo study on the potential vascular relaxation activity of the methanolic extracts of OPL, papaya shoots, lemongrass, and green chili using larger conductance vessels (aortic ring) and smaller resistance vessels (mesenteric vascular bed), which were isolated from four-month-old male Wistar–Kyoto rats [44]. These vessels were pre-contracted with noradrenaline. Among the four extracts that were tested, the OPLE exerted the greatest vascular relaxation in both the endothelium-intact aortic rings (84%) and the mesenteric vascular bed (70%). These values were slightly higher than those reported for acetylcholine (80% and 60%, respectively), which was the positive control (Appendix A). Therefore, it can be inferred that the OPLE is a more effective vascular relaxant compared to acetylcholine. The removal of the endothelium from the aortic ring or the inhibition of endogenous NO in the endothelium-intact aortic rings with NOLA, an inhibitor of the NOS and a vasoconstrictor, resulted in a complete loss of the vascular relaxation of the OPLE, therefore suggesting that the OPLE’s vascular relaxation was endothelium-dependent [44].

The vascular endothelium (endothelial cells) plays an essential role in controlling vascular tone (e.g., vascular relaxation and contraction) through the release of contractile factors and relaxants such as NO and prostacyclin [49,50]. Studies have shown that certain flavonoids can mediate the release of the NO and prostacyclin, and therefore, this justifies the considerable (*p* < 0.05) vasorelaxant action of the OPLE in the aortic ring [44,50].

NOLA is known to reduce the relaxant effects of flavonoids. Therefore, the loss in the vascular relaxation activity of the OPLE upon treatment of the endothelium-intact aortic rings with NOLA confirmed that the flavonoids in the OPLE were involved in the release of NO from the endothelium of the aortic rings [50].

#### 3.5.5. Hypocholesterolemic Activity

The in vivo hypocholesterolemic effects of the OPLE were studied by Irine et al. (2003) using four different groups of male New Zealand white rabbits [18]. Irine et al. observed that at week 8, the increase in the serum total cholesterol, HDL-C, and triglyceride levels were significantly lower in the rabbits that were administered with 0.2% OPLE and 1% oil palm leaf powder (OPLP) compared to the rabbits that were on a cholesterol diet (Appendix A). Although both of the OPL diets were effective in attenuating the increase in the levels of the serum total cholesterol, HDL-C, and triglycerides at week 8, the hypocholesterolemic effects of the OPLE and OPLP were weakened upon continuous feeding of the rabbits with 1% cholesterol up to 16 weeks (Appendix A). Several studies have reported that certain hypocholesterolemic agents are effective in rabbits fed with relatively low dietary cholesterol, but they are not effective in rabbits fed with high-cholesterol diets [18].

The weight of the rabbits from the four different groups were also monitored over 16 weeks. While the weight of the rabbits fed with normal diet remained unchanged, the weight of those fed with either the cholesterol diet or the cholesterol diet supplemented with the OPLE displayed an increase in week 8. The OPLE-supplemented diet suppressed the body weight from increasing beyond week 8 (Appendix A) [18].

At the end of 16 weeks, the rabbits were sacrificed, and the weights of the liver, kidney, and heart were recorded (Appendix A). No significant difference was observed in the weights of the kidney and heart in all groups. On the other hand, the livers showed considerably different values in their weights, with those from the rabbits fed with the cholesterol diet being significantly heavier than the rest [18].

The cholesterol diet supplemented with 0.2% OPLE and the cholesterol diet supplemented with 1% OPLP showed no effects on the ALT, GGT, TCK, creatinine, and urea levels. ALT and GGT were used to detect liver dysfunction, while normal kidney function was indicated by normal urea and creatinine concentrations. Therefore, the lack of adverse effects suggested that an OPL diet for 16 weeks was safe for the rabbits’ liver and kidney functions [18].

#### 3.5.6. Neuroprotective/Neurogenesis

The in vivo neuroprotective and neurogenesis effects of ethanolic OPLE in nitric oxide (NO)-deficient male Wistar Kyoto rats were studied by Mohamed et al. (2013) [19]. The neuron viability of these rats was evaluated in three subfields of the hippocampus: CA1, CA3, and DG.

In the NO-deficient group, 53% of the viable pyramidal cells in the CA1 region were destroyed under chronic NO-deficient conditions. Oral administration with OPLE significantly (*p* < 0.001) prevented this neurodegeneration by retaining 74% of the viable neurons (*p* < 0.001). Captopril was less effective in providing neuroprotection in this region, as the viable pyramidal cell count was lower than that of OPLE (Appendix A) [19].

NO deficiency was more pronounced in the CA3 region compared to the CA1 region. Only 20% of the viable neurons remained in the CA3 region in the NO-deficient group. Treatment with OPLE was found to significantly (*p* < 0.001) protect the CA3 pyramidal cells by maintaining the viability at 72%. On the other hand, although less effective, captopril maintained the viability of the pyramidal cells at 47% (Appendix A) [19].

The neurodegeneration in the DG region of the NO-deficient rats was slightly less serious than the CA1 region but more extensive than the CA3 region with only 32% viable granule cells remaining. Once again, the neuroprotective properties of the OPLE (retained 76% of the neurons) were notably more effective (*p* < 0.001) compared to those of captopril (retained 56% of the neurons) (Appendix A) [19].

The normal group supplemented with the OPLE showed a slightly higher viable neuron count than the normal control group, which could have been due to neurogenesis or the protection against natural neuron losses that occur with aging. This suggests the absence of neurotoxicity by the OPLE [19].

#### 3.5.7. Phytoestrogenic Properties/Osteogenic Activity

Bakhsh et al. (2013) investigated the phytoestrogenic properties of the ethanolic OPLE on estrogen deficiency-induced bone loss [14]. Compared to the control group, the OVX rats displayed significantly lower femur and tibia ash weights, lower femur and tibia bone density (fresh and dry masses), lower femur and tibia bone calcium content, and lower T-ALP levels (Appendix A). On the other hand, OVX rats treated with either 150 mg OPL/kg body weight or 300 mg OPL/kg body weight exhibited higher values for all of the aforementioned parameters compared to the control group, the OVX rats, and the 2% green tea supplemented OVX rats (Appendix A) [14].

#### 3.5.8. Fungicidal/Antimicrobial Activity

The fungicidal activity of the methanolic OPLE was evaluated against *Candida albicans* using the disk diffusion assay and the broth microdilution technique by Sreenivasan et al. (2010). The potency of the OPLE was accessed by the presence or absence of inhibition zones, zone diameters, and MIC values. Ciprofloxacin was employed as the standard antibiotic [5]. The results of the disc diffusion assay revealed that at the tested concentrations, the maximum inhibition zone of the OPLE was smaller than that of ciprofloxacin (Appendix A). Based on the broth microdilution technique, the OPLE was found to be a promising anti-fungal agent against *Candida albicans*, as the minimum inhibitory concentration (MIC) was only 6.25 mg/mL [5]. It has been previously reported that extracts with MIC values below 8 mg/mL are considered to possess effective antimicrobial activity [5].

An in vivo study of the fungicidal activity of the OPLE was conducted using Sprague–Dawley strain albino rats. Full-thickness wounds were made on the shaved dorsal area of the Sprague–Dawley strain albino rats and inoculated with 0.1 mL *Candida albicans* (approximately 10^9^ CFU). The wounds were treated topically with 10% formulated crude OPLE, while the control rats were treated with only soft yellow paraffin for 16 days. Skin tissues were collected from the control and the OPLE-treated groups. Histological analysis was carried out on these tissues. Results showed that there were more clusters of the *Candida albicans* in the control group compared to the OPLE-treated rats (Appendix A) [5].

The antimicrobial activity of the OPLE was determined by using microbial count analysis. Granulated tissues from both the treated and control groups were excised prior to the application of the OPLE formulation on days 4, 8, 12, and 16 and were subsequently examined. The microbial count of the OPLE-treated rats on day 16 was notably lower (10² CFU/g tissue) compared to the control group (10⁴ CFU/g tissue) (Appendix A) [5].

#### 3.5.9. Antihypertensive Properties and Cardiovascular Effects

Jaffri et al. (2011) evaluated the antihypertensive properties and cardiovascular effects of methanolic OPLE on hypertensive rats that were _L_-NAME induced, NO-deficient, and normotensive male Wistar–Kyoto rats [31]. Hypertensive rats fed with captopril or OPLE significantly attenuated the BP increase and returned it to normal values. Catechin-rich OPLE was known to reduce systolic and diastolic blood pressure [51]. Treatment with OPLE was noted to significantly reduce the MDA and SOD activities (*p* < 0.05) in the heart of hypertensive rats, compared to the control and _L_-NAME groups. No significant differences were observed in the CAT and GPx activities among all the groups. Induced NO deficiency significantly increased myocardial fibrosis (60%) compared to the control group. Neither OPLE nor captopril administration could attenuate this damage. In terms of myocardium, the _L_-NAME fed with captopril group demonstrated near normal myocardium. The _L_-NAME fed with the OPLE group had no significant effect on influencing the thickness of the myocardium.

Regarding the changes in the coronary arteries, the hypertensive rats’ group had significantly thicker coronary artery walls (*p* < 0.005) compared to the control group. The concomitant administration of OPLE or captopril decreased wall-to-lumen ratios to normal values in the hypertensive rats’ group. The attenuated BP increase helped to minimize the artery structural changes caused by hypertension [31].

The concomitant administration of the OPLE or captopril had no effects on the normotensive rats’ group. _L_-NAME-induced hypertension causes the overproduction of local angiotensin II in the heart and aorta. This leads to increased vascular O_2_ formation through the expression of NADPH-dependent oxidase in the aortic smooth muscle cells. _L_-NAME inhibits NO synthesis, impairs endothelial-dependent vasodilation, and induces hypertension due to excess O_2_ formation [52]. Thus, NADPH oxidase and endothelial NO synthase are the main contributing sources of ROS [53]. The polyphenols (flavonoids e.g., catechin) in the OPLE were able to attenuate BP increase by quenching the ROS [54]. It is also important to note that the OPLE did not alter the BP of normotensive rats [31].

OPLE was able to attenuate hypertension but not the associated cardiac hypertrophy. Similar results were reported in another study conducted by Pechánová et al. (2004) with catechin-rich red wine. BP was reported to be reduced along with increased NO synthase activity. This prevented myocardial fibrosis in _L_-NAME rats without affecting the left ventricle hypertrophy [53].

#### 3.5.10. Cytotoxicity/Toxicity Effects

Cell viability assessment is essential in ensuring that the effect of a particular treatment is not affected by the toxicity of the samples that are being investigated. The MTT assay, an in vitro cytotoxicity assay, conducted by Che et al. (2020) revealed that the crude OPLE, the flavonoid-enriched fractions, the flavonoid *C*-glycoside-enriched fractions, and the purified flavonoid *C*-glycosides did not exert any toxicity effects to the 3T3 fibroblast cells [21]. This was evident at the tested concentrations (1.563–25 µg/mL), as indicated by a cell viability percentage of above 79–155% after 48 h of incubation (Appendix A) [21]. Another study conducted by Jaffri et al. (2011), determined that throughout the experimental duration on normotensive and hypertensive Wistar–Kyoto rats, the OPLE caused no hypotensive or other apparent toxic effects in the normotensive rats [31]. It was shown by the normal appearance of the myocardium and unchanged antioxidant status. Hence, OPLE was not cardiotoxic.

In a separate study by Ibraheem et al. (2012), the ethanolic OPLE was assessed for its in vivo subacute and acute oral toxicities on Sprague–Dawley female rats [23]. The oral administration of OPLE showed no significant changes in the tested rats’ behavior, respiration, and neuronal response. There were also no noticeable effects in the liver and renal functions and in the metabolic and hematologic status. The OPLE was considered to be safe with its LD_50_ value below 5 g/kg (Appendix A). As described by Ecobichon (1997), the failure of any test compounds in producing adverse effects at a dose limit that exceeds 5 g/kg is considered practically non-toxic [55]. Thus, the toxicity effect of the OPLE can be categorized as no observable adverse effect level (NOAEL) [23].

### 3.6. Strengths and Limitations

The results from this review were summarized and synthesized from a pool of 16 individual studies that investigated the flavonoid composition and the pharmacological properties of the OPLE. This systematic review provides a comprehensive overview of the existing research from 2001 to 2021. Thus, the results summarized within this systematic review provide a broad representation of the pharmacological activities of the OPLE studies up to date. This review follows the Preferred Reporting Items for Systematic Reviews and Meta-Analysis (PRISMA) and provides an in-depth and systematic summary of existing research [15]. Several restrictions and drawbacks were identified with this systematic review. Despite the wide variation in animal models across the different studies included in this review, it should be noted that a large majority of studies included in this review shared similar animal models of either Wistar rats or Sprague–Dawley rats. Hence, this systematic review provides a good homogeneity of experimental method as well as a relatively accurate and comprehensive summary of the latest available literature. Lastly, this systematic review has also addressed a vital gap in the current literature surrounding the pharmacological activities of the OPLE. This systematic review is not without limitations. There exist several heterogeneities in this study: for example, the solvent systems, methods used for the extraction of the OPL, and the lack of consistency in the units of measurement. Except for the antioxidant properties, there is a noticeable lack of analysis on the other pharmacological properties. The language selection was limited to English, which may have introduced a language bias. As stated by Morrison et al. (2012), languages other than English should be included to minimize the risk of a biased summary effect [56]. Clinical trials on humans (in vivo) were not included in this systematic review. Moreover, the purpose of a systematic review is to provide an overview of the existing evidence and not critically appraise the quality of the studies. Hence, this systematic review only synthesized a comprehensive overview of the existing evidence based on the collected studies, factoring out their intrinsic qualities.

### 3.7. Future Research

Although the OPLE have been scientifically reported to exhibit a broad spectrum of pharmacological properties, the secondary metabolites that were responsible for these therapeutic effects are still unknown. Therefore, research needs to be carried out on a large scale of plant material to isolate a sufficient amount of the major as well as the minor chemical constituents to explore their pharmacological activities and mechanisms of action.

It has now been established that the OPLE has promising antioxidative potential in several assays; however, as there is still a lack of studies of its other pharmacological properties besides antioxidant activity, these should be the focus of future research. The dose-dependent characteristics of the pharmacological properties of the OPLE was described by several studies. Ergo, an extensive study would be recommended for the identification of its optimal concentration. There are also synergistic effects of the phenolic compounds in the OPL reported in the literature retrieved, and this gap should be further identified and explored. By far, there are only two studies that assessed the cytotoxicity/toxicity effects of the OPLE. Even though it was categorized as practically non-toxic, the limit should be identified. Future research should be conducted using a consistent concentration, solvent system, and units of measurement, to allow easier generalization and understanding.

## 4. Conclusions

This review provides scientific justification for the potential development of the OPLE as pharmaceutical as well as cosmeceutical products. Overall, this review thoroughly discusses and describes the prospect of the pharmacological properties of the leaves of *E. guineensis*. Moreover, it also described in depth the flavonoids that were present in the leaves of *E. guineensis*, which included the catechins, luteolin, vitexin, isovitexin, orientin, and isoorientin. These flavonoids could be responsible for the broad spectrum of pharmacological activities exhibited by the leaves of *E. guineensis*: antioxidant, wound healing, hypoglycemic, vascular relaxation, hypocholesterolemic, neuroprotective, neurogenesis, phytoestrogenic, osteogenic, fungicidal, antimicrobial, antihypertensive, cytotoxicity, and toxicity.

## Figures and Tables

**Figure 1 pharmaceuticals-14-00961-f001:**
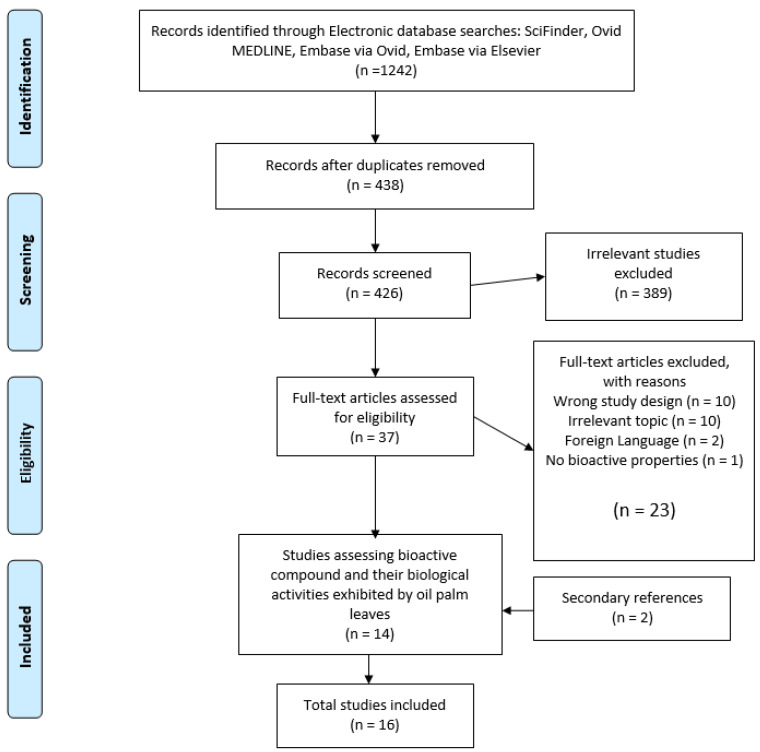
PRISMA flow diagram for article selection process.

**Figure 2 pharmaceuticals-14-00961-f002:**
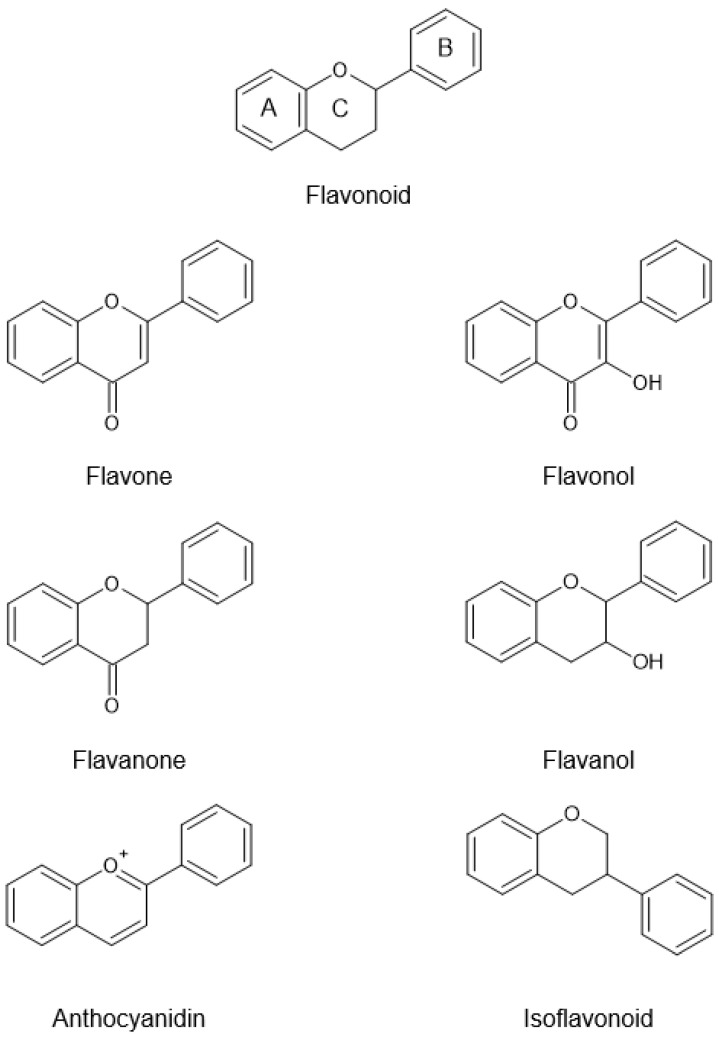
Subclasses of flavonoids.

**Figure 3 pharmaceuticals-14-00961-f003:**
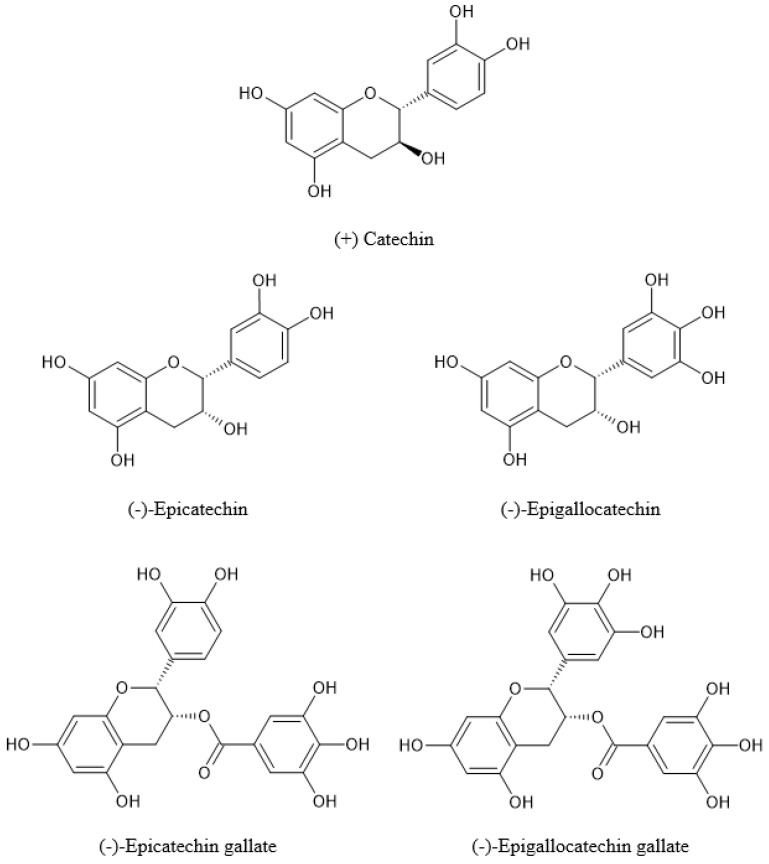
Structure of the main types of catechins found in the leaves of *E. guineensis*.

**Figure 4 pharmaceuticals-14-00961-f004:**
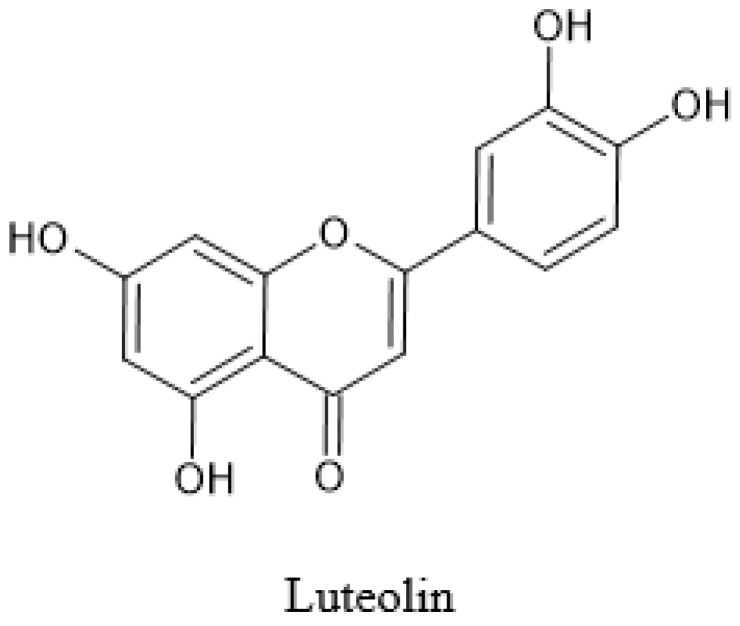
Structure of luteolin.

**Figure 5 pharmaceuticals-14-00961-f005:**
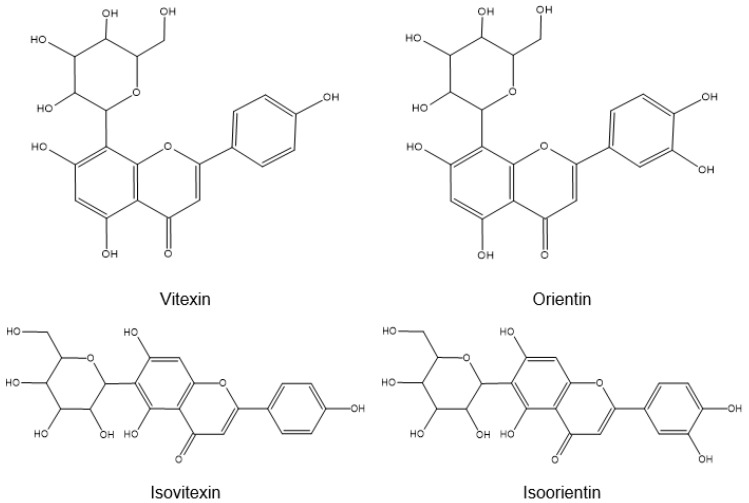
Structure of vitexin, isovitexin, orientin, and isoorientin.

**Table 1 pharmaceuticals-14-00961-t001:** Keywords used for search strategy.

	Keywords
Oil Palm Leaves	“*Elaeis guineensis*” OR “Oil Palm Leaves” OR “Oil Palm Leaf” OR “Oil Palm Frond” OR “Oil Palm Fronds”
Bioactive Compounds	“Phytochemicals” OR “Phytochemical” OR “Phytonutrients” OR “Phytonutrient” OR “Flavonoids” OR “Flavonoid” OR “Chalcones” OR “Chalcone” OR “Flavones” OR “Flavone” OR “Isoflavones” OR “Isoflavone” OR “Flavanols” OR “Flavanol” OR “Flavanones” OR “Flavanone” OR “Flavonols” OR “Flavonol” OR “Anthocyanidins” OR “Anthocyanidin” OR “Phenolics” OR “Phenolic” OR “Phenolic acids” OR “Phenolic acid” OR “Coumarins” OR “Coumarin” OR “Lignins” OR “Lignin”

**Table 2 pharmaceuticals-14-00961-t002:** Detection and identification of flavonoids in the leaves of *E. guineensis* and their TPC and TFC values.

Reference	Methods for Detecting and Identifying Flavonoids	Types of Flavonoids	TPC (mg GAE g^−1^ Sample or Extract)	TFC
[2]	N/R	N/R	INSOL fraction: 118.44 ± 0.0 9BHT (positive control): 116.22 ± 0.04 MeOH extract: 63.67 ± 0.14 WATER extract: 61.13 ± 0.28 EA extract: 44.03 ± 0.03 HEX extract: 16.96 ± 0.13	N/R
[4]	FTIR Spectroscopy	Flavonoids	Moderately high phenolic content of the aqueous methanolic extract: 0.33	N/R
[18]	HPLC	Luteolin	MeOH extract: 24.3 ± 1.7 Green text extract: 22.5 ± 1.7	N/R
[19]	HPLC	(−)-Epigallocatechin, (+)-catechin, (−)-epicatechin, (−)-epigallocatechin gallate, (−)-epicatechin gallate, and their glucosides	N/R	N/R
[20]	N/R	N/R	Extract prepared from dried leaves: 10.2 Extract prepared from wet leaves: 5.1	N/R
[11]	N/R	N/R	OPAL M1: 1.160 ± 0.001 OPAL M2: 0.982 ± 0.001 OPAL M3 0.304 ± 0.044 OPAL M4: 0.440 ± 0.006 Classification refers to Appendix A.	OPAL M1: 1.16 ± 0.001 ^c^ OPAL M2: 0.55 ± 0.002 ^c^ OPAL M3 0.30 ± 0.002 ^c^ OPAL M4: 0.33 ± 0.001 ^c^
[21]	UHPLC-UV/PDA LCMS/MS	Major compounds: orientin, isoorientin, vitexin, isovitexin Minor compounds: luteolin-6-8-di-*C*-hexose, apigenin-6,8-di-*C*-hexose, luteolin-6-8-di-*C*-hexose, apigenin-6-*C*-pentose-8-*C*-hexose, apigenin-6-*C*-hexose-8-*C*-pentose, luteolin-6-*C*-hexose-8-*C*-deoxyhexose, and apigenin-6-*C*-hexose-8-*C*-deoxyhexose.	N/R	N/R
[22]	NMR Spectroscopy	(+)-Catechin, vitexin, isovitexin, orientin, isoorientin, and (−)-epicatechin	Aqueous methanol extract: 393.61 ± 36.11 Absolute methanol extract: 213.08 ± 41.61 Ethyl acetate-methanol extract: 174.19 ± 32.40 Ethyl acetate extract: 121.71 ± 32.78	Aqueous methanol extract: 129.72 ± 8.70 ^b^ Absolute methanol extract: 135.40 ± 9.76 ^b^ Ethyl acetate-methanol extract: 121.48 ± 6.67 ^b^ Ethyl acetate extract: 5.94 ± 5.38 ^b^
[23]	Phytochemical qualitative analysis	Flavonoids	Ethanol extract: 52.4	Ethanol extract: 15.4 ^a^
[5]	Phytochemical qualitative analysis	Flavonoids	N/R	N/R

TPC: Total Phenolic Content; TFC: Total Flavonoid Content. Superscript: ^a^ indicates (mg CE g^−1^ extract). ^b^ indicates (mg QCE g^−1^ extract). ^c^ indicates (mg ECE g^−1^ sample).

## Data Availability

Data sharing not applicable.

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
