# Peer review of "Flavonoid Composition and Pharmacological Properties of Elaeis guineensis Jacq. Leaf Extracts: A Systematic Review"

_pharmaceuticals, 2021, doi:10.3390/ph14100961_

Round 1

Reviewer 1 Report

This review provides a synthetic view od the beneficial antioxidant  effects of Palm biomass. This review is well written,

My only comment confers to bibliographie thre are some references which are missing or reference is not correctly situated in the text, and some years are missing.

It was being sometimes difficult to follow the references with all the errors with the hyperlinks that should be corrected.

Author Response

Reviewer 1

Comments

Reply

My only comment confers to the bibliographie There are some references which are missing or reference is not correctly situated in the text, and some years are missing. It was being sometimes difficult to follow the references with all the errors with the hyperlinks that should be corrected.

References have been amended according to the comments given by the reviewer.

Changes have been made according to the comments given by the reviewer. Changes have been highlighted in green

Reviewer 2 Report

Tow et al. conducted a systematic review on the flavonoids present in Elaeis guineensis leaf extracts and their pharmacological effects comprising antioxidant, hypoglycemic, vascular, hypocholesterolemic etc. activities. The flavonoid pattern is thoroughly presented. The reference list consists of 56 altogether entries.

The whole manuscript is written in an excellent and fluent style. There are no spelling mistakes.

Unfortunately, the processing of the manuscript is not in a final form, thus the numeration of the pages and the lines has to be improved by the EDITORIAL OFFICE. Therefore the manuscript is available in two parts, comprising part A with pages 1/25 to 7/25 (lines 1-116) including the chapters 1-3 and part B with pages 1/25 to 18/25 (lines 1-759) starting with chapter 3.1.

Regarding the manuscript there are only some minor issues which need to be addressed:

Part A:

Line 30: the correct name of the plant is „Elaeis oleifera (Kunth) Cortés“ (i.e. the accent on the „e“ of Cortés should be added)

Line 34: „… tree are widely used …“

Table 1: Why did you include the terms „phenolic acid“ / „phenolic acids“, „coumarins“ / „coumarin“, „lignins“ / „lignin“ as keywords as these natural classes do not belong to the group of flavonoids? On the other side why did you not use the terms „catechin“ / „catechins“ and also „epicatechin“/ „epigallogatechin“ as keywords for your search as these compounds are mentioned in chapter 3.2.1 (flavan-3-ols) and are also depicted in figure 3?

Figure 1 and Line 104-107: I could not understand how you reduced the 1242 records to 438. In line 104 to 107, you mention the number of 640 and also the number of 453. Please insert these numbers also in the scheme (Figure 1) in order to explain the selection of the studies to the readership.

Table 3: Please explain „N/R“ in the list of abbreviations!

Table 3 / Study by Che et al., 2020b: Regarding the major compounds isoorientin is mentioned. However, this compound is also mentioned as a minor compound. Please check again!

Part B (starting with chapter 3.1!)

Line 21/22: „… prenylated derivatives. Flavonoid glycosides …“ (full stop is missing)

Line 35-37: this is true for the leaves of E. guineensis, but not for other plants. My proposal: „The structure of these catechins in E. guineensis leaves is characterized …“ Please check again!

Line 98: Please explain „OPAL“ in the list of abbreviations!

Lines 481-500: I propose to use the same way of citation throughout the manuscript. Please check again the citation for references 32, 27, 18 and 55!

In conclusion the systematic review is well perfomed and the manuscript is very well prepared and is of great interest for the international audience. Only a few minor corrections and additions are needed which are easily performed. This publication is an excellent basis for the development of new health products.

Author Response

Reviewer 2

Comments

Reply

Line 30: the correct name of the plant is „Elaeis oleifera (Kunth) Cortés“ (i.e. the accent on the „e“ of Cortés should be added)

“Cortes” has been amended to “Cortés”. The correction has been highlighted in yellow. Kindly refer to line 30 on page 1 of the manuscript

Line 34: „… tree are widely used …“

“Is” has been changed to “are” and highlighted in yellow. Kindly refer to line 34 on page 1 of the manuscript

Table 1:

Why did you include the terms „phenolic acid“ / „phenolic acids“, „coumarins“ / „coumarin“, „lignins“ / „lignin“ as keywords as these natural classes do not belong to the group of flavonoids?

On the other side why did you not use the terms „catechin“ / „catechins“ and also „epicatechin“/ „epigallogatechin“ as keywords for your search as these compounds are mentioned in chapter 3.2.1 (flavan-3-ols) and are also depicted in figure 3?

The original plan was to include in the present review the various types of phenolic compounds and flavonoids which have been identified in the oil palm leaves between 2001-2020. Later on, my co-authors and I decided to focus on the flavonoid composition of the oil palm leaves instead.

We did not include “catechin“ / “catechins“/ “epicatechin“ / “epigallogatechin” as the keywords for the search as these keywords would be too specific and therefore would limit the scope of the search. Instead, we decided to use the various classes of flavonoids as the keywords to conduct a wider search in order to gather as much information as possible on the various types of flavonoids which have been previously identified in the oil palm leaves

Figure 1 and Line 104-107: I could not understand how you reduced the 1242 records to 438. In line 104 to 107, you mention the number of 640 and also the number of 453. Please insert these numbers also in the scheme (Figure 1) in order to explain the selection of the studies to the readership.

The paragraph was amended for a clearer explanation. The search yielded a total 1242 articles from all the electronic databases. 438 articles remained after removing duplicates were removed from Endnote. The remaining articles were then uploaded onto Covidence, and the duplicates were removed again by Covidence, with 426 articles remaining. Kindly refer to lines 104-107 which have been highlighted in yellow under section 3 on page 4 of the manuscript

Table 3: Please explain „N/R“ in the list of abbreviations!

The abbreviation has been added to the list of abbreviations and highlighted in yellow. Kindly refer to page 15 of the manuscript

Table 3 / Study by Che et al., 2020b: Regarding the major compounds isoorientin is mentioned. However, this compound is also mentioned as a minor compound. Please check again!

Isoorientin was accidentally included in the category of “minor compounds”. It has now been removed. Kindly refer to Table 2 (the section which has been highlighted in yellow) on page 6 of the manuscript

Line 21/22: „… prenylated derivatives. Flavonoid glycosides …“ (full stop is missing)

The full stop has been added after “prenylated derivative” and has been highlighted in yellow. Kindly refer to lines 20-21 under section 3.2 on page 2 of the manuscript

Line 35-37: this is true for the leaves of E. guineensis, but not for other plants. My proposal: „The structure of these catechins in E. guineensis leaves is characterized …“ Please check again!

The sentence has been corrected (highlighted in yellow) according to the suggestion given by the reviewer. Kindly refer to lines 34-37 under section 3.2.1 on page 2 of the manuscript

Line 98: Please explain „OPAL“ in the list of abbreviations!

OPAL refers to the oil palm leaves alcoholic extract. The explanation (highlighted in yellow) has been included in the list of abbreviations on page 15 of the manuscript

Lines 481-500: I propose to use the same way of citation throughout the manuscript. Please check again the citation for references 32, 27, 18 and 55!

The citations have been amended throughout the entire manuscript